# Histone Deacetylase Inhibitors for Peripheral T-Cell Lymphomas

**DOI:** 10.3390/cancers16193359

**Published:** 2024-09-30

**Authors:** Ruxandra Irimia, Pier Paolo Piccaluga

**Affiliations:** 1Department of Hematology, “Carol Davila” University of Medicine and Pharmacy, 030167 Bucharest, Romania; ruxandra.irimia@gmail.com; 2Johns Hopkins Bloomberg School of Public Health, Baltimore, MD 21205, USA; 3Department of Medical and Surgical Sciences, School of Medicine, University of Bologna, 40138 Bologna, Italy; 4Biobank of Research, IRCCS Azienda Ospedaliera-Universitaria di Bologna, Institute of Hematology and Medical Oncology “L&A Seràgnoli”, 40138 Bologna, Italy

**Keywords:** peripheral T-cell lymphoma, histone deacetylase (HDAC), HDAC inhibitor (HDACi), targeted therapy, epigenetic, DNA mutation, TET2, DNMT3A

## Abstract

**Simple Summary:**

Nodal peripheral T-cell lymphomas (PTCLs) are among the most aggressive lymphomas and in the majority of cases are considered incurable. Therefore, there is a cogent need for novel therapies. In this review, the authors analyzed the biological rationale, the experimental experience, and the real-world data concerning the use of histone deacetylase inhibitors (HDACis) to treat PTCLs.

**Abstract:**

Histone deacetylase inhibitors (HDACis) are being recognized as a potentially effective treatment approach for peripheral T-cell lymphomas (PTCLs), a heterogeneous group of aggressive malignancies with an unfavorable prognosis. Recent evidence has shown that HDACis are effective in treating PTCL, especially in cases where the disease has relapsed or is resistant to conventional treatments. Several clinical trials have demonstrated that HDACis, such as romidepsin and belinostat, can elicit long-lasting positive outcomes in individuals with PTCLs, either when used alone or in conjunction with conventional chemotherapy. They exert their anti-tumor effects by regulating gene expression through the inhibition of histone deacetylases, which leads to cell cycle arrest, induction of programmed cell death, and,the transformation of cancerous T cells, as demonstrated by gene expression profile studies. Importantly, besides clinical trials, real-world evidence indicated that the utilization of HDACis presents a significant and beneficial treatment choice for PTCLs. However, although HDACis showed potential effectiveness, they could not cure most patients. Therefore, new combinations with conventional drugs as well as new targeted agents are under investigation.

## 1. Introduction

Peripheral T-cell lymphomas (PTCLs) constitute a relatively rare and heterogeneous group of non-Hodgkin lymphomas (NHLs) that arise from post-thymic T cells. According to the most recent classification of lymphoid neoplasms, PTCLs encompass a series of different entities that demonstrate various and intricate clinicopathologic features [1]. The most commonly observed subtypes are the so-called PTCL not otherwise specified (PTCL NOS), the T-follicular helper-related PTCL, angioimmunoblastic type (AITL) and the anaplastic large cell lymphomas (ALCL), and anaplastic kinase-negative (ALK−) or positive (ALK+).

Overall, except for a few more indolent forms arising in the gastrointestinal tract, the skin, or the bone marrow, the majority of PTCLs demonstrate highly aggressive behavior [2]. Currently, the most commonly used first-line therapy is represented by CHOP (cyclophosphamide, doxorubicin, vincristine, and prednisone) or CHOEP (cyclophosphamide, doxorubicin, vincristine, etoposide, and prednisone). Some studies indicated that fit, chemosensitive patients may benefit from front-line autologous stem cell transplant (ASCT) [2]. In contrast, despite the aggressive nature of this disease, allogeneic SCT is still considered an experimental option in this setting.

In patients with relapsed/refractory (R/R) peripheral T-cell lymphoma (PTCL), the median progression-free survival (PFS) and median overall survival (OS) were reported to be inferior to 5 and 6 months, respectively [2,3,4]. Based on the poor outcomes, biologically meaningful targeted therapies are urgently warranted. In the last few years, a series of new agents have been approved in this setting, including pralatrexate, brentuximab vedotin (for CD30-positive cases), and histone deacetylase (HDAC) inhibitors (HDACis) [5]. As far as the latter is concerned, HDACs play a crucial role in gene expression regulation, and can impede the growth of tumor cells by triggering apoptosis and subsequent cell death [6].

In the following, the authors summarize the rationale for HDACis in TCL as well as the main clinical evidence from clinical trials.

## 2. HDAC in Cellular Biology

Epigenetic regulation of gene expression by acetylation and deacetylation of histones represents an important mechanism involved in cellular homeostasis, metabolism, division, and apoptosis [7]. Alterations in the finely balanced processes of acetylation (driven by histone acetyltransferases) and deacetylation (led by histone deacetylases) ultimately lead to abnormal DNA transcription, genomic instability, and epigenetic disease [8].

HDACs primarily regulate transcription by removing acetyl groups from the e-amino groups of histone tail lysine residues [9,10]. This contributes to chromatin condensation, thus limiting the accessibility of transcription factors to DNA.As transcriptional corepressors, HDACs cause the nucleosome to compact, resulting in gene suppression. In addition to histone substrates, HDACs also regulate the stability and function of non-histone proteins by post-translational deacetylation [11,12,13]. This process is another important mechanism for regulating both physiologic and pathologic cellular processes, including gene transcription, signal transmission, protein folding, autophagy, DNA repair, cell proliferation, and metabolism (Figure 1) [11,12,13]. The non-histone substrates of HDACs include transcription factors that have been traditionally associated with various malignancies, such as nuclear factor B (NF-B), TP53, GATA1, GATA2, STAT3, BCL6, and heat shock protein 90 (HSP90), regulatory proteins, but also chaperone proteins, structural proteins, and steroid receptors [9,10].

In mammals, 18 types of HDACs have been classified into four classes based on sequence patterns, cellular localization, tissue specialization, and enzymatic activity (Table 1).

Class I, HDACs 1, 2, 3, and 8, are ubiquitously expressed in human tissues and are mainly located in the nucleus, where they histone acetylation. Recently, it has been discovered that HDAC8 is also extensively found in the smooth muscles, where it modulates contractility [15,16]. While HDAC1 and 2 are solely expressed at the nuclear level,HDAC3 can also be found in the cytoplasm and at the cell membrane level. However, during mitotic advancement, HDAC3 is exclusively localized on the mitotic spindle where it preserves correct kinetochore–microtubule attachment and chromosome alignment [17].

HDAC2 frequently associates with HDAC1 within co-repressor complexes such as SIN3, NURD, MiDAC, and CoREST. In addition to this role, HDAC2 overexpression has been shown to regulate IFNγ signaling and contributes to the nuclear translocation of PDL1 [16]. Meanwhile, HDAC3 plays a critical role in the metabolic regulation by promoting fatty acid oxidation. Beyond metabolism, HDAC3 is essential for the inflammatory response, particularly in the host defense against bacterial infections, as it enhances TNF-mediated NF-kB activation [18].

The class I HDACs contribute to cancer biology through an array of mechanisms. One of the main substrates of HDAC1 is represented by the p53 protein. The *TP53* gene plays a pivotal role in maintaining genomic stability by regulating the cell cycle, particularly at the G1/S and G2/M checkpoints, and induces apoptosis in response to severe DNA damage [19]. The main effector of the p53 function is p21 [19]. P21 expression is induced additionally in a p53-independent fashion, through acetylation of it’s coding regions [20]. The class 1 HDACs have a profound effect on the regulation of these processes. Under normal circumstances, HDAC1 and p53 have balanced antagonistic effects. DNA damage leads to a location shift between HDAC1 and the activated p53 at the promoter region of *TP21*, which induces *TP21* transcription, followed by cell apoptosis [20]. The activity of p53 itself is also regulated by acetylation and deacetylation, with HDACs 1, 2, and 3 serving as repressors [21]. Mutations in the *TP53 genes* correlate with loss of cell cycle regulation and have been described in a variety of solid and hematological malignancies [19]. Also, overexpression of HDAC1, 2, and 3 has been documented in numerous malignancies, such as pancreatic, colon, and breast cancers [21,22]. HDAC1 and HDAC2 specifically contribute to the maintenance of mutant p53 function and promote MYC recruitment in various pancreatic carcinoma cell lines, while HDAC8 has been shown to sustain and activate mutant *TP53* through the transcription factors HoxA5 and YY1 in breast cancer models [22,23].

Another target protein for the class 1 HDACs with a role in malignant transformation is BCL6. BCL6 is an essential transcription factor required for the normal germinal center maturation of the B lymphocytes and a key determinant of lymphomagenesis [24,25]. The function of BCL6 is closely linked to the activity of histone deacetylases [26]. Under physiological conditions, BCL6 activity is negatively regulated by acetylation. However, HDAC2 overexpression has been observed in various B-cell lymphomas, where it directly contributes to the abnormal activation of BCL6, thereby promoting uncontrolled tumor growth [8,27].

HDAC8 is also overexpressed in various adult and pediatric cancers; however, the mechanisms by which it supports malignant progression are convoluted, still poorly understood, and tumor-specific [28]. The inhibition of HDAC8 induces apoptosis through various pathways such as activation of the p53/p21 pathway, Bcl-2-mediated apoptosis, and caspase-dependent apoptosis [29,30].

The transcription factor c-Myc is an additional target of the class 1 HDACS. C-Myc is over-expressed in various aggressive B-cell malignancies such as Burkitt lymphoma, diffuse large B-cell lymphoma, or mantle cell lymphoma [31]. In addition to serving as a transcription factor, c-Myc regulates the expression of miRNAs, some of which code for tumor-suppressor genes [32]. Several studies have shown that Myc regulates the expression of the tumor-repressor miRNAs by recruiting HDAC3 and impairs their activity through altered acetylation [32].

In addition to their impact on the regulation of various transcription factors, the class 1 HDACs are also involved in another pivotal process associated with cancer progression, angiogenesis. The main regulator of angiogenesis is hypoxia factor 1 alpha (HIF-1 α) via the secretion of pro-angiogenic cytokines [33]. HDAC1 induces the overexpression of HIF-1 α independently by decreasing the expression of p53 and von Hippel–Lindau factor and, therefore, promotes tumor-associated angiogenesis [34].

Class II HDACs were further classified into IIa and IIb subclasses due to differences in sequence homology and domain organization [9]. Subclass IIa (HDAC4, 5, 7, and 9) exhibits signal-dependent nucleocytoplasmic shuttling as well as tissue-specific expression [9,35]. Subclass IIb (HDAC6 and 10) contains two catalytic HDAC domains and is localized in the cytoplasm in a restricted number of tissue types [9]. HDAC6 is widely known for deacetylating particular cytosolic non-histone substrates involved in tumor genesis, progression, and metastasis. Tubulin, cortactin, peroxiredoxin, HSP90, and heat shock transcription factor-1 (HSF1) are examples of frequent substrates [36]. HDAC6 plays an intricate role in the initiation and progression of malignant transformation. In vitro studies of various tumoral cell lines have suggested that the oncogenic potential of *Ras* mutations is highly dependent on the activity of the HDAC6 and the knockdown of HDAC6 correlates with impaired tumoral growth and decreased clonogenic potential both in vitro and in vivo [37]. Interestingly, the elevated expression of HDAC6 in the tumor microenvironment also plays a critical role in sustaining an immunosuppressive state, which supports tumoral growth and progression in ovarian and breast cancer [38,39]. Furthermore, HDAC6 transcription is driven by estrogen stimulation, establishing a connection between this histone deacetylase and estrogen-dependent tumors [40]. In contrast to HDAC6, which is an acetyl lysine deacetylase, HDAC10 is a N8-acetylspermidine deacetylase and has been linked to dysregulated polyamine metabolism and neoplastic disorders such as colon cancer, prostate cancer, and neuroblastoma [41]. HDAC10 enhances cell survival via autophagy in response to chemotherapeutic treatments and pathogen infection, according to several studies [41,42]. As a result, inhibition of HDAC10 autophagy may be a unique method for sustaining chemotherapeutic efficacy, particularly in the treatment of advanced-stage neuroblastoma [41]. However, HDAC10 expression seems to correlate with a favorable outcome in some types of cancers, such as pulmonary tumors. The sequencing of tumoral samples from lung cancer patients has shown a strong correlation between the expression of class II HDACs, particularly a low HDAC10 expression, and prognosis [43]. The tumor-suppressor activity of HDAC10 in lung cancer appears to be connected to the KRAS pathway [43,44]. The deletion of HDAC10 is also associated with maintenance of an immunosuppressive environment, characterized by high density of the tumor-supporting M2 macrophages [45].

HDACs class III are structurally distinct compared to classes I and II and are collectively called Sirtuins. They are generally involved in metabolism regulation, and unlike other HDAC classes, the deacetylase activity of Sirtuins is dependent on NAD and consequently on theNAD/NADH ratio. Currently, seven HDACs class III have been described, and the roles they play range from normal cellular homeostasis to aging, inflammation, and oxidative stress [46].

Sirtuins too play a significant role in cancer biology. Among them, SIRT1 is the most extensively studied [47]. SIRT1 is often upregulated in cancer cells, and it contributes to tumorigenesis by various mechanisms such as alteration in the acetylation of transcription factors such as p53, DNA-repairing enzymes, or signaling cytokines [47]. However, data on the exact role of SIRT1 in cancer remain contradictory, with some studies suggesting context-specific effects that include both pro-tumorigenic and tumor-suppressive actions [48]. One of the most important substrates of class III HDACs, with a clear role in oncogenesis, is c-Myc [31]. The interrelation between cMyc and HDACs is bidirectional [49]. It has been demonstrated that the Myc proteins upregulate the expression of SIRT2 in neuroblastoma and pancreatic cancer. Consequently, SIRT2 represses the Myc protein degradation pathway mediated through NEDD4, leading to a vicious circle that maintains an over-expression of the Myc oncoproteins.

Because of a unique sequence pattern, HDAC11 is the sole class IV HDAC representative [9,50]. HDAC11 is a promising therapeutic target in chronic metabolic disorders as it exhibits both a deacetylase activity as well as an efficient defatty-acylase activity [50,51].

HDAC11 is overexpressed in several carcinomas, including prostate, breast, and ovarian cancers [52]. However, subsequent extensive sequencing across 33 cancer types suggests a context-dependent ambivalent effect of HDAC11, a high expression being correlated with improved prognosis in renal carcinomas, glioma, and rectal adenocarcinoma [53]. The exact role of HDAC11 in tumor progression is currently still under debate; however, it seems that there are significant differences depending on the tumor type. In hepatocellular carcinoma, HDAC11 promotes cancer stemness and progression by regulating glycolysis through the LKB1/AMPK signaling pathway [54]. At the same time, in pituitary carcinomas, HDAC11 expression deacetylates the p53 transcription factor HEY1, which subsequently negatively correlates with p53 expression [55]. In contrast, in colorectal cancer, HDAC11 has a positive effect as it prevents metastasis through the inhibition of the matrix metalloproteinase 3 [56].

## 3. HDACi as Anti-Cancer Treatment

HDACs are commonly overexpressed in malignant cells, and the use of inhibitors can cause proliferation arrest, differentiation, and apoptosis in many cancers [57]. Over five decades have passed since the emergence of the first HDAC inhibitor as a therapeutic agent. HDAC inhibitors, such as valproic acid, have first been used in neurology and psychiatry as mood stabilizers and anti-epileptics, but their efficacy started being tested in malignant disorders shortly afterward [58,59,60,61]. Sodium butyrate and trichostatin A (TSA), the classic and representative HDAC inhibitors, are among the first compounds used experimentally on cancer cells [58,59]. Sodium butyrate, a natural short-chain fatty acid produced by bacteria in the colon, was the first compound that sparked the interest in using HDAC inhibitors in the treatment of cancer due to its therapeutic success in a limited number of acute leukemia cases [62]. Similarly, the administration of TSA, a compound derived from a streptomyces metabolic product, led to the differentiation of leukemia cell lines as well as neoplastic cells apoptosis; however, the poor biodisponibility made it impossible to use in vivo [59].

Shortly after, a number of prodrugs were developed, but the action of these compounds was non-specific, had a short effect, harbored substantial side effects, and obtaining effective inhibitory concentrations in vivo was often unattainable [63] The interest in HDAC inhibitors led however in the past two decades to the development of more effective and less toxic compounds, such as SAHA (vorinostat), Depsipeptides (such as romidepsin), Belinostat (PXD-101), Panobinostat (LBH-589), and Entinostat (MS-275). These drugs have shown efficacy both in vitro and in vivo and vorinostat, belinostat, panobinostat, and romidepsin were granted FDA approval for human use.

The low expression of tumor-supressor cells in human malignancies has been directly linked to an alteration in the acetylation processes [64]. In addition, in vitro studies and phase I and II clinical trials prove that small molecular HDAC inhibitors (HDACis) are highly efficient in up-regulating tumor-suppressor gene expression, decreasing tumor growth, and triggering programmed cell death [62,65]. The anticancer mechanism of HDAC inhibitor is therefore based on three main mechanisms: (1) cell cycle arrest and apoptosis, (2) inducing cell differentiation, and (3) prevention of angiogenesis [62].

In the past years, the interest in using HDACis in various hematologic malignancies has been rekindled by a better understanding of the targetable molecular mechanisms and the availability of more potent and safer compounds [65]. In acute myeloid leukemia, the selective inhibitors of HDAC1 and HDAC2 have demonstrated synergistic effects when combined with azacitidine, both in in vivo as well as in vitro models [66]. Panobinostat has also shown robust antileukemic responses in AML models, particularly in t(8;21) AML, by degradation of the AML1/ETO fusion protein and subsequent terminal myeloid differentiation [67]. The success of these interventions led to the exploration of potential novel HDAC inhibitors with anti-AML activity both as single agents and in combination with the Bcl-2 inhibitor Venetoclax [68,69].

In Hodgkin lymphoma, HDAC inhibitors are being investigated for their potential to overcome drug resistance and immune escape mechanisms. While HDAC inhibitors alone may not be sufficient for treating relapsed or refractory classical Hodgkin lymphoma, they show a synergistic effect in combination with PD-1 inhibitors [70]. The combination of vorinostat and pembrolizumab showed very promising responses in this population, with an overall response rate of 100%, a complete response rate of 44%, a 6-month progression-free survival (PFS) rate of 80%, and a 6-month overall survival (OS) rate of 100% [71]. In diffuse large B-cell lymphoma, HDAC inhibitors like chidamide have demonstrated cytotoxic effects on DLBCL cells by impacting key survival pathways such as PI3K/AKT and mTOR signaling [72]. Furthermore, the combination of HDAC inhibitors with other agents, such as the proteasome inhibitor bortezomib, has shown synergistic effects in DLBCL preclinical models [73]. In myelodysplastic syndromes, treatment with HDACis in monotherapy has demonstrated limited clinical activity [74]. However, their efficacy seems to be potentiated by the combination of azacitidine and decitabine, and this synergistic combination could potentially represent a therapeutic option for high-risk MDS patients, ineligible for allo-HCT [75,76].

In multiple myeloma, the pan-HDAC inhibitor panobinostat has received FDA approval for the treatment of relapsed/refractory patients, in combination with bortezomib [77,78]. Panobinostat targets class I, II, and IV histone deacetylases, resulting in gene expression modulation that promotes cell cycle arrest, induces apoptosis, and suppresses angiogenesis [78]. Panobinostat drives apoptosis in myeloma cells by regulation of the pro-apoptotic factors p21, caspase-3/7, and PARPs, while simultaneously reducing the intracellular level of anti-apoptotic proteins such as Bcl-2 and Bcl-XL [79,80]. Additionally, Panobinostat inhibits HDAC6, which disrupts protein homeostasis and induces the degradation of proteins essential for myeloma cell survival [79,80].

## 4. Biological Rationale for HDACi Use in Peripheral T-Cell Lymphomas

PTCL development is a multistep process involving the accumulation of genetic and epigenetic changes that target both the post thymic T cell intrinsically, as well as the tumoral environment [81,82]. Gene expression profiling studies demonstrated molecular subtypes of PTCLs that closely resemble their normal T-cell counterparts, revealing the possible cell of origin. For instance, angioimmunoblastic T-cell lymphoma and related subtypes have a genetic signature that resembles normal T follicular helper cells [82,83,84,85,86]. Likewise, PTCL, not otherwise specified, includes two main molecular subgroups associated with Tbet-expression (Th1-like) and GATA3-expression (Th2-like) [82,83,84,85,86]. The pathogenesis of PTCL is driven by a combination of altered mechanisms such as the TCR/CD3, Notch, Jak/STAT, or RHOA pathway [81,83]. PTCLs are also characterized by various epigenetic alterations that involve mainly TET2, DNMT3A, and IDH2 [81,83]. In addition to these intrinsic alterations, the tumor microenvironment also plays a crucial role in PTCL pathogenesis. The interaction between neoplastic T cells and the surrounding stromal cells supports tumor growth, survival, and disease progression [81,87].

A prominent contributor to the genomic instability and aberrant gene expression that characterizes PTCLs as well as other types of cancer is the dysregulation of acetylation and deacetylation [88,89]. Among malignancies, PTCLs included, HDACs play a significant role in regulatingdownstream gene expression networks and signaling pathways, primarily through their deacetylation activity on transcription factors and signaling mediators.

Gene expression profiling (GEP) studies, followed by mutational analyses, provided sufficient evidence for considering HDACis a rational approach for mature TCL. [84,88,90,91,92,93,94]. HDACis can prompt apoptosis in PTCLs by addressing aberrant signaling pathways governed by HDACs, arising from distorted gene expression or premature degradation of pro-apoptotic proteins [57]. For example, *TET2* and *DNMT3A* gene mutations are common in the early stages of PTCL pathogenesis, and HDAC1 and 2 mediate TET2 protein deacetylation and degradation via the ubiquitin–proteasome pathway [89]. Also, HDAC1, HDAC2, and HDAC3 play a regulatory role in restricting the transcription of STAT3 target genes within another commonly altered pathway, the JAK/STAT pathway, resulting in subsequent epigenetic silencing of tumor-suppressor genes [9,90]. Inhibition of these HDACs with small molecules can thus lead to cell growth arrest or apoptosis [9,90].

Alterations in the *TP53* gene and/or disruptions in the p53 pathway are also involved in the dysregulation of cell cycle arrest and apoptosis in PTCLs [90]. In T-cell lymphomas, *TP53* mutations manifest in late stages and are indicative of poor prognosis [90]. The deacetylation of *TP53* by HDAC1 inhibits its activity, reduces the expression of the p21 protein, and diminishes apoptotic signaling [20,21]. HDAC inhibitors-induced acetylation can in turn increase the expression of p53 and restore the response to therapy in *TP53-*deleted tumoral cells [90].

The deleterious effects of HDACs in PTCLs also include the NF-κB activation in the TCR/CD3 pathway, which can be mitigated by inhibiting HDAC3 using small molecules such as Vorinostat [89,92]. Panobinostat also showed promising results in this setting by increasing the acetylation of HSP90, repressing the expression of the anti-apoptotic protein BCL2, and downregulating mitogen-activated protein kinase pathway signaling [84,93].

Besides apoptosis, autophagy is another possible therapeutic mechanism of action of HDAC inhibitors in T-cell lymphomas. One example is SAHA (vorinostat), which promotes autophagy by inhibiting mTOR and augmenting the efficacy of the autophagic factor L3 [94,95].

Another mechanism by which HDAC inhibitors can interfere with T-cell leukemogenesis is the alteration of cytokine signaling. Interleukins 2, 4, 7, and 15 are recognized as promoters of disease, and the use of HDACis disrupts the production of interleukins by the tumoral environment and malignant cells. One conclusive example is the use of arginine butyrate, which induces cell apoptosis in peripheral T-cell lymphoma by altering the Il-2 environmental signaling [58].

An additional important therapeutic effect of HDAC inhibitors in PTCLs is the modulation of the supporting tumoral microenvironment by inhibiting angiogenesis. Various studies have shown that the administration of HDACis alters the pro-angiogenic signaling pathways by reducing the expression of proangiogenic genes, including basic fibroblast growth factor, vascular endothelial growth factor, angiopoietin, and endothelial nitric oxide synthase [93].

### 4.1. Rationale for HDAC Inhibitors in Virus-Induced PTCL

The role of viral infection in the etiopathogenesis of lymphomas has been extensively described. In concern to T-cell lymphomas, the Epstein–Barr virus (EBV) and Human T-cell lymphotropic virus type 1 (HTLV-1) are recognized as the main viral drivers of malignant transformation [81].

EBV is a gamma-herpesvirus that is thought to play a role in the progression of several human cancers, including lymphoma, gastric carcinoma, and nasopharyngeal carcinoma [96]. In the latest edition of the WHO classification, EBV-positive TCLs represent a distinct group of entities that includes EBV-positive nodal T- and NK-cell lymphoma and extranodal NK/T-cell lymphoma [1]. In PTCLs, EBV has been detected at varying frequencies, ranging from 20% in PTCL-NOS to 30–100% in extranodal NK/T-cell lymphoma, nasal type. This association is linked to a poorer prognosis [97,98]. In PTCLs, EBV can infect either neoplastic T cells (as in some PTCL/NOS and the above-mentioned EBV-positive TCL) or bystander B cells, (as in AITL), implying that EBV plays both a direct and indirect role in PTCL pathogenesis [81].

Romidepsin, a class I HDACi, stimulates the EBV lytic cycle and facilitates apoptosis when associated with ganciclovir in Burkitt lymphoma by inhibiting HDAC1, 2, and 3 and upregulating p21 [99]. EBV latent protein LMP1 upregulates *STAT5A* and recruits HDAC1/2 to the CEBPA gene locus, which is involved in neoplastic plasticity control and cellular dedifferentiation [100]. In this case, HDACis (romidepsin and chidamide) have been shown to restore CEBPA expression and reverse cellular dedifferentiation in EBV+ NPC in vitro [100].

HTLV-1 infection is associated with virtually all cases of adult T-cell leukemia/lymphoma (ATLL) and represents the major driver of malignant transformation [82]. In ATLL, the oncoprotein Tax physically and functionally interacts with HDAC1 [101]. Interestingly, HDACi can regulate Tax expression and therefore induce viral protein expression, enhancing T-cytotoxic response toward HTLV1 infected cells [102]. However, whether this can be useful as anti-ATLL treatment has to be demonstrated.

### 4.2. HDACi as Therapeutic Option in Mature T-Cell Lymphomas

Based on early clinical results in some acute leukemias and B-cell lymphomas and the preclinical evidence of their potential efficacy, HDACis were extensively tested in PTCLS.

#### 4.2.1. HDACis in First-Line Treatment of PTCL Patients

Due to the relative scarcity of these lymphomas, there is no supporting data coming from randomized clinical trials to guide the initial therapeutic approach. However, it is generally accepted that the standard initial therapy consists of CHOP or CHOP-like regimens followed by autologous stem-cell transplantation in eligible patients. The first major breakthrough in the treatment of newly diagnosed PTCL patients was the development of brentuximab vedotin, and the ECHELON study proved the efficacy of the BV-CHP combination compared to the standard CHOP regimen (Table 2) [5].

In the past years, the incorporation of HDACis in the treatment of newly diagnosed PTCL patients has been explored. Two studies evaluated the combination of romidepsin with CHOP either as a phase I/II or phase III clinical trial [110]. In the phase III trial, the combination romidepsin–CHOP failed to prove superiority compared to CHOP in terms of PFS (12.0 mo (95% CI, 9.0–25.8) vs. 10.2 mo (95% CI, 7.4–13.2)), HR 0.81 (95% CI, 0.63–1.04; *p =* 0.096), ORR (63% vs. 60%), or OS (51.8 mo (95% CI, 35.7–72.6) vs. 42.9 mo (95% CI, 29.9–not evaluable)) [110]. Belinostat in combination with standard CHOP has also been explored in this setting. The data shows that the combination provided an ORR of 81% with a rate of CR of 71% [105]. Consolidation with autologous stem cell transplant was not considered in these studies.

Another agent utilized in combination with the two standard chemotherapic regimens for untreated PTCL patients was chidamide. Chidamide in combination with CHOP provided an 89.3% ORR with 57.1% of the participants attaining a CR [106]. When combined with CHOEP, chidamide conferred a 60.2% ORR, with a complete response rate of 40.7%; however, the addition of etoposide did not offer a benefit in terms of event-free survival for the PCTL except for the ALK + ALCL [108].

One additional study also evaluated the safety and efficacy of romidepsin in combination with the hypomethylating agent azacytidine, a combination that proved to be highly efficacious in the treatment-naive population. The reported ORR was 70% with a CR of 50% [109]. In this study, autologous stem cell transplant was allowed and 4 patients received the procedure, with 3 still being in remission at the time of the report (median PFS 20.8 months) [109].

The efficacy of HDACi-based combination therapy for untreated PTCL patients was recently revised in terms of ORR, CR rate, and PR rate in a meta-analysis comprising a pooled total of 502 patients, from 7 clinical trials [111]. The pooled ORR was 72%. One of the seven studies used belinostat-based treatment and involved 21 patients, yielding an ORR of 86% [111]. Three studies employed chidamide-based treatment and comprised a total of 224 patients, with a pooled ORR of 74% [106,108,111]. Three studies with a total of 257 individuals used romidepsin-based treatment, resulting in a pooled ORR of 64% [103,104,109,111].

The pooled CR rate in the analysis was 44% [111]. However, after subgroup analysis, the CR rate for the belinostat-based therapy was 67%, the pooled CR rate in the romidepsin-based therapy subgroup was 43%, and for chidamide-based treatment, it was 42% without statistically significant differences between the three agents (*p* = 0.07) [111].

The current therapeutic approaches for newly diagnosed PTCL patients could benefit from the addition of novel agents. While various HDAC inhibitors combined with CHOP have shown promising overall response rates, none have consistently demonstrated superiority over CHOP alone in terms of progression-free survival or overall survival. 

#### 4.2.2. HDACi in Relapsed/Refractory PTCL Patients

The treatment of R/R PTCL is challenging as the therapeutic options are limited and the response to conventional chemotherapy poor. Along with brentuximab vedotin and the dihydrofolate reductase inhibitor pralatrexate, the HDAC inhibitors represent the most prominent newly developed class of drugs for this category of patients [84,91,92,93]. To date, four HDAC inhibitors have been approved for the treatment of relapsed PTCL—class I, II, and IV inhibitors (Vorinostat (SAHA), Belinostat), the selective HDAC1, 2, 3, and 10 inhibitor chidamide, and the selective class I inhibitor Romidepsin. Among these, only Romidepsin, Belinostat, and Vorinostat have received FDA approval for the treatment of R/R PTCL [112].

The reported ORR of single-agent HDAC inhibitor therapy in this setting is 24% for Vorinostat, 25% for Romidepsin, 25.8% for Belinostat, and 28% for Chidamine [112,113,114]. However, real-world multicentric data reported response rates as high as 33% in the R/R PTCL population when treated with a romidepsin single agent [115].

One of the most promising combinations in this setting is represented by the association of hypomethylating agents with HDAC inhibitors. The real-life experience for the combination of romidepsin with azacytidine reported an ORR of 76.9% with a CR rate of 53%, significantly higher than the 61% and 48% reported in the clinical trial setting [116].

A recent meta-analysis assessed pooled data from 16 studies on HDACi for R/R PTCL [111]. These studies included overall 662 relapsed/refractory patients and used HDAC inhibitors either in monotherapy or in therapeutic combinations with hypomethylating agents, conventional chemotherapy, and proteasome inhibitors +/− lenalidomide [111]. Most of the studies evaluated romidepsin which received FDA approval in 2007 for the treatment of previously treated PTCLs. The overall response rate was 37%; specifically, the ORR in the HDAC inhibitor-based combination therapy cohort was 45% (95% CI, 36–54%), whereas the ORR in the HDAC inhibitor monotherapy subgroup was 33% (95% CI, 27–38%), a difference that reached statistical significance difference (*p* = 0.02) [111]. There were no differences in terms of efficacy when different HDACis were used (romidepsin, belinostat, or chidamide). The overall pooled CR rate was 14% with a CR rate of 22% in the combination treatment subgroup and 13% in the monotherapy subgroup (*p* = 0.02). There was no statistical difference in the CR rate between the three types of HDAC inhibitors (*p* = 0.06) [101].

The treatment of relapsed/refractory PTCL remains challenging, with conventional chemotherapy yielding only modest responses. As a result, HDAC inhibitors have emerged as one of the most promising alternatives. Although their efficacy as single agents is limited, combining them with hypomethylating agents has shown significantly improved response rates, particularly in real-world settings.

#### 4.2.3. HDACi Efficacy across PTCL Subtypes

Three of the studies on untreated PTCLs did include subtype analyses [103,107,108]. All 196 patients had an ORR of 68% (95% CI, 62–75%, fixed effect model). In the PTCL/NOS, AITL, and ALK-neg ALCL categories, the pooled ORR was 58%, 71%, and 76%, respectively. However, no significant difference (*p* = 0.17) was determined [111].

The histological subtype analysis in R/R PTCL patients included eight investigations [111]. Pooled together, the 418 patients had an ORR of 32%; the ORR in PTCL-NOS patients was 29%, in ALK-negative ALCL was 27% (95% CI, 16–38%), while in the AITL subgroup was 44%, strongly indicating a more efficient response compared to other subgroups (*p* = 0.01). The selective efficacy of HDAC inhibitors in AITL was further confirmed by real-life retrospective data from a national Israeli study [115].

## 5. Real-World Experience with HDAC Inhibitors in PTCL

To evaluate the real-world impact of HDACis in PTCL, we conducted a systematic literature review. A comprehensive review using Scopus, PubMed, and EMBASE was conducted for English-language articles published from 1 January 2000 through 30 November 2023. Key search terms included peripheral T cell lymphoma, Romidepsin, Chidamide, Vorinostat, Belinostat, real world, case report, and HDAC inhibitors. Exclusion criteria included case reports with less than 3 patients and non-English texts. One thousand fifty-three papers have been identified, out of which only five matched the inclusion criteria.

The first study by Shi et al. was conducted in China and involved 256 patients with relapsed or refractory PTCL, treated with chidamide either in monotherapy or in combination with conventional chemotherapy [117]. For patients receiving chidamide in monotherapy, the overall response rate (ORR) was reported to be 39.06%, while the combination with chemotherapy ensured an ORR of 51.18%. Notably, subgroup analyses within the study included patients with AITL and ALK+ anaplastic large cell lymphoma (ALK+ ALCL). In AITL, chidamide monotherapy demonstrated an ORR of 49.23%, which increased to 71.43% when combined with chemotherapy, emphasizing potential subtype-specific responses. For ALK+ ALCL, the ORR was even more promising at 66.67% when treated with chidamide monotherapy and 100% when chidamide was combined with chemotherapy. Overall, the PFS for chidamide monotherapy was 129 days (95% CI 82 to 194) vs. 152 days (95% CI 93 to 201) for the combination chidamide + chemotherapy (*p* = 0.3266). In the monotherapy group, the AITL patients achieved a median PFS of 144.5 days, while the combination therapy delivered an impressive median PFS of 176 days [117]. When evaluating the efficacy of various chemotherapy regimens in association with the HDAC inhibitor, CHOP-like regimens were the most effective, with a prolonged PFS of 172 days, while the platinum-containing regimens resulted in a PFS of only 119 days [117].

Regarding the safety profile, the most common adverse effects experienced by the monotherapy group are hematological (anemia, thrombocytopenia, neutropenia) and nausea/vomiting.

In a separate study by Shimony et al. conducted in Israel, romidepsin monotherapy was administered to 42 PTCL patients with relapsed or refractory disease, resulting in an ORR of 33% including a 12.5% rate of CR [115]. The lower response rate suggests that the efficacy of HDAC inhibitors may vary among different agents, although of note, this study population contained a relapsed-refractory category of patients who received a median of 2 prior lines of therapy (65% received 2 or 3 previous lines of therapy, range 1–5). However, the duration of response exceeded 1 year (13.4 months), with a median OS of 7.1 months,although the PFS was only 2.2 months. In alignment with the findings from the other studies, AITL subtype was associated with a longer EFS [115].

The most common side effects were hematological, although one-third of the patients required hospitalization due to infectious complications [115].

The investigation by Kalac et al. which included patients from the USA and Australia explored the combination of romidepsin and azacitidine in 26 PTCL patients, including 3 newly diagnosed patients. The results showed an impressive ORR of 76.9% with a complete response rate (CRR) of 53% and a PFS of 13.3 months [116]. As expected, the ATL patients benefited from a 69.5% ORR, with a CRR of 60.8%. Notably, this study included patients undergoing stem cell transplantation, indicating the potential role of ASCT transplantation in this treatment setting [116]. From a mutational standpoint, the TET2, IDH, and DNMT3A mutated cases benefited most from this combination therapy with an ORR of 77.7% [116].

Median PFS for the 26 patients was 13.3 months and the OS was not reached. Notably, the median PFS for patients undergoing stem cell transplant was not reached, while for the transplant-ineligible patients, the PFS was 7.07 months [116].

Overall, the combination was well tolerated, the most important side effects including nausea, fatigue, rash, neutropenia, and thrombocytopenia [116].

Another significant contribution comes from Liu et al. in China who conducted a large-scale study involving 548 relapsed or refractory PTCL patients [118]. Chidamide monotherapy showed an ORR of 58.6%, with a 21.1% CR, while the combination with chemotherapy resulted in a higher overall response rate of 73.2% and 25.4% CR, suggesting a potential synergistic effect between HDAC inhibitors and chemotherapy [118]. Furthermore, in AITL patients, chidamide in monotherapy or incombination therapy achieved a remarkable ORRs of 75.1%. The most common side effects of chidamide monotherapy included anemia, thrombocytopenia, neutropenia, and fatigue in one-third of the patients while fatigue was reported by almost 60% of the patients receiving chemotherapy–chidamide combination [118].

The study by Wei et al. in China retrospectively explored the efficacy of chidamide in combination with CHOEP in newly diagnosed patients with PTCL, compared to standard CHOEP therapy [119]. The study included 33 patients in each arm. In this setting, the ORR was 68.8%, with an impressive CRR of 56.3% for the chidamide-CHOEP combination. However, the authors concluded that patients receiving chidamide–CHOEP therapy did not achieve a statistically significant difference compared to CHOEP (ORR 68.8% versus54.8%, *p* = 0.256) although the CRR was double in the combination therapy compared to standard CHOEP (56.3% vs. 25.8%, *p* = 0.014). The median PFS for chidamide–CHOEP was 12 months compared to 7 months in the standard chemotherapy arm, *p* = 0.905, and the OS was 57 months, compared to 30 months, *p* = 0.359. About 30% of the patients in the combination therapy group received stem cell transplant consolidation and 36.4% chidamide maintenance [119].

Notably, a flat PFS curve was obtained in two circumstances: in patients receiving ASCT consolidation as well as in the ones that benefited from chidamide maintenance [119].

In general, the C-CHOEP regimen exhibited good tolerance, with no significantly severe hematologic and non-hematologic toxicities observed when compared to the CHOEP regimen. Hematologic toxicities were the most frequently reported adverse events in both groups [119].

Lastly, Guo et al. investigated chidamide maintenance therapy in 48 transplant-ineligible PTCL patients who received a first-line induction with conventional chemotherapy [120]. The study schema included a 2-cycle consolidation with chidamide + original induction therapy for patients achieving at least a PR after induction, or chidamide + conventional second-line therapy as salvage therapy for patients not achieving a response. For patients who responded, the consolidation was followed by chidamide monotherapy maintenance. The HDAC inhibitor consolidation and maintenance demonstrated an impressive ORR of 93.8% with a CRR of 60.4% [120].

AITL and ALCL (ALK + and −) subtypes presented with a staggering 100% ORR and with CRR of 61.9% and 100%, respectively (ALCL ALK+) [120].

PFS showed no significant distinction between patients achieving CR and PR. In contrast, notable differences were evident when comparing patients undergoing first-line maintenance (40 cases) and salvage therapy (8 cases), with 1-year PFS rates of 80.8% and 46.9% and 2-year PFS rates of 71.9% and 46.9%, respectively (*p* = 0.012). The median PFS was not reached in the first-line maintenance group, while it was 10.1 months in the salvage therapy group [120].

The most common and severe side effects were hematological; however, only one patient experienced infectious complications [120].

Although there is a marked heterogeneity between the reported outcomes in the identified studies, all of them reported superior ORR in the angioimmunoblastic T-cell lymphoma subtype suggesting that AITL may be particularly responsive to HDAC inhibitor therapy compared to other peripheral T-cell lymphoma subtypes. This observation suggests a possible “Achille’s heel” of AITL that makes it more susceptible to the mechanisms of action of HDAC inhibitors. The main studies on real-world experience with HDACi in PTCLs are summarized in Table 3.

## 6. Conclusions

The treatment of PTCL remains challenging as there is still a limited number of available therapeutic agents. In recent years, HDACs have emerged as new epigenetic therapeutic agents for both newly diagnosed as well as relapsed/refractory cases. For the latter case, this is particularly relevant since these patients represent an unmet medical need. The standard therapeutic approach with salvage chemotherapy and stem cell transplant confers modest results for these patients. The biological rationale for employing HDAC inhibitors in PTCLs is based on the existence of dysregulated acetylation and deacetylation processes, which contribute to genomic instability and aberrant gene expression, especially in the relapsed setting. The complex therapeutic mechanisms of HDAC inhibitors in PTCLs target cell cycle arrest and apoptosis, cell differentiation, inhibition of angiogenesis, and modulation of cytokine signaling. These mechanisms target critical components of PTCL’s molecular complexity, offering a comprehensive approach to curbing tumor growth and progression.

HDAC inhibitors have shown efficacy both as monotherapy and in combination regimens, delivering robust response rates with manageable toxicity. In the relapsed/refractory setting, HDAC inhibitor-based therapies appear to be a particularly promising strategy, when combined with standard chemotherapy or with hypomethylating agents. . Intriguingly, the combination with hypomethylating agents such as azacitidine and decitabine appeared to be also strikingly effective in relapsed/refractory patients, prompting further studies testing the dual epigenetic therapy combination.Despite the absence of large randomized trials, these unprecedented response rates are driving a shift in the treatment paradigm. 

Interestingly, although in the relapsed setting, PTCL histology does not seem to significantly impact outcomes when treated with standard chemotherapy, angioimmunoblastic T-cell lymphoma, known for its poor survival rates, appears to benefit most from HDAC inhibitor-based therapies among nodal PTCL subtypes.

On the other hand, it is still hard to predict the subcategory of patients who would benefit most from the incorporation of HDAC inhibitors in the therapeutic scheme. The presence of single genetic lesions (including those affecting epigenetics) did not seem to correlate with clinical response, and reliable predictive biomarkers are therefore lacking. We believe that identifying reliable biomarkers by introducing mutational analysis, gene and miRNA expression is a key step for the future development of HDACi-based therapies in PTCL patients.

## Figures and Tables

**Figure 1 cancers-16-03359-f001:**
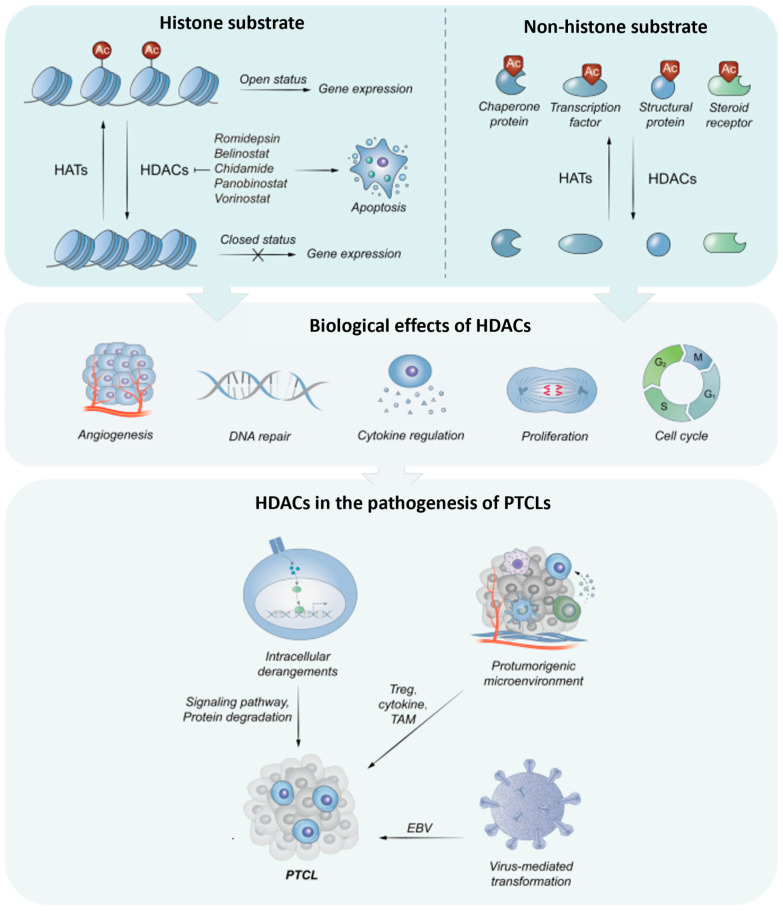
The histone or non-histone substrates and the integrated biological effects of HDACs. The acetylation of histone substrates modulates the chromatin structure to reduce the accessibility to transcriptional regulatory proteins and subsequent gene expression. For non-histone substrates, HDACs have an impact on their activity by acetylating. In general, HDACs contribute to proliferative effects (Adapted from Lu et al. [14]).

**Table 1 cancers-16-03359-t001:** HDAC classes, general characteristics and role in disease.

HDAC	Class	Cellular Localization	Substrate Specificity	Substrates	Function	Expression Pattern	Associated Diseases
HDAC1	I	Nuclear	Histone proteins	Androgen receptor, SHP, TP53, MyoD, SMC4, E2F1, STAT3	Gene regulation, cell cycle control	Ubiquitous	Cancer, neurodegenerative disorders
HDAC2	I	Nuclear	Histone proteins	Glucocorticoid receptor, YY1, BCL6, STAT3	Gene regulation, cell cycle control	Ubiquitous	Cancer, neurodegenerative disorders
HDAC3	I	Nuclear, cytoplasmic, membrane	Histone proteins	SHP, YY1, GATA1, RELA, STAT3, MEF2D	Gene regulation, cell cycle control	Ubiquitous	Cancer, metabolic diseases
HDAC4	II A	Nuclear, cytoplasmic	Histone and nonhistone	GCMA, GATA1, HP1	Muscle differentiation, development	Tissue-specific (heart, skeletal muscle, brain)	Muscular disorders, neurodegenerative diseases
HDAC5	II A	Nuclear, cytoplasmic	Histone and nonhistone	GCMA, SMAD7, HP1	Muscle differentiation, development	Tissue-specific (heart, skeletal muscle, brain)	Muscular disorders, neurodegenerative diseases
HDAC6	IIB	Cytoplasmic	Cytoplasmic proteins	α-Tubulin, HSP90, SHP, SMAD7	Aggresome formation, protein degradation	Tissue-specific (heart, liver, kidney)	Neurodegenerative disorders, cancer
HDAC7	II A	Nuclear	Histone and nonhistone	PLAG1, PLAG2	Vascular development, immune response	Tissue-specific (endothelium, heart, skeletal muscle, pancreas, placenta, thymus)	Cardiovascular diseases, cancer
HDAC8	I	Nuclear	Histone proteins	-	Cell cycle	Ubiquitous	Cancer
HDAC9	II A	Nuclear	Histone and nonhistone	-	Development, cardiac function	Tissue-specific (brain, heart, skeletal muscle)	Cardiovascular diseases, cancer
HDAC 10	II B	Cytoplasmic, nuclear	Cytoplasmic proteins	-	Cellular proliferation, apoptosis	Tissue-specific (liver, spleen, kidney)	Cancer, neurodegenerative diseases
HDAC 11	IV	Nuclear	Histone proteins	-	Gene regulation	Tissue-specific (brain, heart, kidney, testis)	Cancer, inflammatory diseases
SIRT1	III	Nuclear, cytoplasmic	Histone and nonhistone, NAD-dependent	-	Metabolic, stress response	Ubiquitous	Aging-related diseases, metabolism disorders
SIRT2	III	Cytoplasmic, nuclear	Histone and nonhistone, NAD-dependent	-	Cell cycle, homeostasis	Ubiquitous	Metabolic diseases, cancer
SIRT3	III	Mitochondrial	NAD-dependent	-	Mitochondrial function, energy metabolism	Ubiquitous	Metabolic diseases, cancer
SIRT4	III	Mitochondrial	NAD-dependent	-	Metabolism	Tissue-specific (pancreas)	Metabolic diseases, cancer
SIRT5	III	Mitochondrial	NAD-dependent	-	Metabolism	Ubiquitous	Metabolic diseases, cancer
SIRT6	III	Nuclear	Histone and nonhistone, NAD-dependent	-	DNA repair, genome stability	Ubiquitous	Aging-related disease, cancer
SIRT7	III	Nucleolar	Histone and nonhistone, NAD-dependent	-	Ribosomal DNA transcription, cell growth	Ubiquitous	Cancer

**Table 2 cancers-16-03359-t002:** Clinical trials on HDACi in PTCLs.

Study	Design	Treatment	ASCT Permitted	Subtype Efficacy	Toxicity Profile (Grade ¾ Events)
Dupuis, 2015 [103]	Phase I/II single-arm	Ro-CHOP	No	N/A	Cardiac toxicity Febrile neutropenia Hematologic toxicities
Bachy, 2022 [104]	Phase III, RCT	Ro-CHOP	No	AITL (PFS 19.5 mo vs. 10.6 mo)	>10% difference in ≥grade 3 hematologic toxicities
Johnston, 2021 [105]	Phase I, single arm	Bel-CHOP	No		Hematologic toxicity Febrile Neutropenia Nausea SAE rate 43%
Guy, 2021 [106]	Phase I, single arm	Chidamide-CHOP	No	N/A	Hematologic toxicity Vomiting
Wang, 2022 [107]	Phase II, single arm	Chidamide+ Prednison+ Etoposide + Thalidomide	No	AITL (90.2%/ 54.9%)	Hematologic toxicity
Zhang, 2021 [108]	Phase Ib/II, single arm	Chidamide + CHOEP	No	ALK+ AITL (65.9%/41.5%)	Hematologic toxicity
Falchi, 2021 [109]	Phase II, single arm	R-Azacytidine	Yes (4 patients/3 in remission)	AITL (80%/60%)	Hematologic toxicity

N/A—non available.

**Table 3 cancers-16-03359-t003:** Real-world experience with HDACi in PTCLs.

Article	Country of Experience	Number of Patients	Subtype of PTCL	Therapeutic Approach	Stem Cell Transplant Included	Results ORR
Shi, Y. [117]	China	256	PTCL	Chidamide monotherapy	No	39.06%
Shi, Y. [117]	China	32	AITL	Chidamide monotherapy	No	49.23%
Shi, Y. [117]	China	13	ALK+ ALCL	Chidamide monotherapy	No	66.67%
Shi, Y. [117]	China	127	ALK+ ALCL	Chidamide + Chemotherapy	No	51.18%
Shimony, S. [115]	Israel	42	PTCL	Romidepsin monotherapy	No	33%
Kalac, M. [116]	USA, Australia	26	PTCL	Romidepsin–azacitidine	Yes (1 Allo, 7 auto)	76.9%
Kalac, M. [116]	USA, Australia	19	AITL	Romidepsin–azacitidine	Yes (1 Allo, 7 auto)	69.5%
Liu, W. [118]	China	261	PTCL	Chidamine monotherapy	No	58.6%
Liu, W. [118]	China	287	PTCL	Chidamide + Chemotherapy	No	73.2%
Liu, W. [118]	China	177	AITL	Chidamine monotherapy/Chidamide + Chemotherapy	No	75.1%
Wei, C. [119]	China	32	PTCL	Chidamide + CHOEP	Yes	68.8%
Guo, W. [120]	China	48	PTCL	Chidamide maintenance	No	93.8%

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
