# Peer review of "Histone Deacetylase Inhibitors for Peripheral T-Cell Lymphomas"

_cancers, 2024, doi:10.3390/cancers16193359_

Round 1

Reviewer 1 Report

Comments and Suggestions for Authors

This review summarized the studies on the HDACi in the treatment for peripheral T-cell lymphomas and present a bright perspective. 

I have some suggestions:

The manuscript needs to be better organized. 

1. The narration of Session 2 can be more interesting if it can be more closely related with cancer biology, since the authors already summarized the characteristics of all the types of HDAC inhibitors in Table 1. 

2. Examples of HDACi use in cancer treatment can be presented in Session 3, for example, use of HDACi for AML treatment and glioblastoma treatment. The authors can talk more about the use of HDACi in cancers other than PTCLs.

3. For Session 4, I suggest the authors introduce the pathogenesis of PTCLs briefly, then introduce the target or pathway in the pathogenesis the HDACi can work.

4. For Session 4.2, the evidences for the HDACi in PTCLs treatment needs to be better organized and the authors can summarize and comment briefly following each category of evidences. 

Author Response

Dear Sir/Madam,

Thank you very much for taking your time to review our manuscript and help us improve our work. In response to your valuable suggestions, we have made extensive changes in the manuscript and addressed all your commnets as follows:

  1. The narration of Session 2 can be more interesting if it can be more closely related with cancer biology, since the authors already summarized the characteristics of all the types of HDAC inhibitors in Table 1. -We have included more information regarding the involvement of various HDACS, from each class, in the pathogenesis of various cancers.
  2. Examples of HDACi use in cancer treatment can be presented in Session 3, for example, use of HDACi for AML treatment and glioblastoma treatment. The authors can talk more about the use of HDACi in cancers other than PTCLs. – We have added an extensive paragraph about the usage of HIDAC inhibitors in various malignancies based on your siggestion
  3. For Session 4, I suggest the authors introduce the pathogenesis of PTCLs briefly, then introduce the target or pathway in the pathogenesis the HDACi can work. – we have added a paragraph summarizing the pathogenesis of PTCLs
  4. For Session 4.2, the evidences for the HDACi in PTCLs treatment needs to be better organized and the authors can summarize and comment briefly following each category of evidences. – we have reorganized this section, started with the treatment of newly diagnosed PTCLs and then continued with the treatment for relapsed/refractory patients. We have included a brief summary at the end of each of the 4.2 subsections

Thank you again for your valuable comments!

Reviewer 2 Report

Comments and Suggestions for Authors

This is an interesting review article, in which authors disclose initially the role  of HDACi in cancer, therafter the clinical trials and finally the role of this kind of therapy in the real world. 

Although vorinostat and romidepsin have been approved by the FDA, authors should also mention other drugs that have also been approved, including belinostat. 

Additionally on lines 306 &307 authors mention a phase III trials recarding the addition of romidepsin to R-CHOP schema. However, they describe only the ORR and CR with romidepsin, but not wiwth RCHOp. It would be convenient thal also ORR and CR with RCHOP could be added to this manuscript, as well as the HR and confidence interval. 

I have none other comments.

Author Response

Dear Sir/Madam,

Thank you very much for taking your time to review our manuscript and help us improve our work.

We have addressed all your comments and made the following changes:

  1. Although vorinostat and romidepsin have been approved by the FDA, authors should also mention other drugs that have also been approved, including belinostat.  – we have updated the information regarding all the approved FDA HIDAC inhibitors (rows 299 – 302).
  2. Additionally on lines 306 &307 authors mention a phase III trials recarding the addition of romidepsin to R-CHOP schema. However, they describe only the ORR and CR with romidepsin, but not wiwth RCHOp. It would be convenient thal also ORR and CR with RCHOP could be added to this manuscript, as well as the HR and confidence interval.  -we have added this information (rows 482 – 486)

Thank you again for your valuable comments!